# Unified Pose Embeddings: Utilizing Euclidean Space for Simplified Topology Alignment

## Abstract

Generative models for human motion synthesis have demonstrated remarkable capabilities across tasks such as text-to-motion generation, motion inbetweening, style transfer, and motion captioning. However, their adoption in industry remains limited, largely due to challenges in data representation. Industry applications often require diverse articulated skeleton topologies tailored to specific use cases, which are further constrained by limited data availability. Existing methods address these challenges by aligning datasets through shared subsets or unified representations. However, these approaches rely on error-prone alignment processes, limiting their flexibility and scalability. In this work, we leverage Euclidean space to represent human poses, bypassing the need for alignment in configuration space and significantly simplifying the learning objective. Using Euclidean space also frees us from the need to use a common subset representation and allows us to represent poses in any complexity we desire. To disentangle pose and body shape, we introduce a simple yet effective learning strategy. Our method achieves robust inverse kinematics with minimal data requirements, needing just over five minutes of motion capture data to integrate new topologies. We demonstrate the effectiveness of our topology-agnostic representation across three downstream tasks: motion retargeting, text-to-motion generation, and motion captioning.

## 1 Introduction

Generative models for human motion synthesis have gained significant traction within the research community, demonstrating their ability to address a wide range of tasks, including text-to-motion generation, motion inbetweening, style transfer, and motion captioning. Despite these advancements, their adoption in industry remains limited. A key challenge contributing to this gap lies in the choice of data representation: industry applications often require unique articulated skeleton topologies tailored to specific use cases and character designs, compounded by limited data availability. Figure 1 shows one such character, while Figure 2 (a)–(c) illustrates the variability in human skeletons for the same pose.

The configuration space of an articulated human pose consists of two components: the skeleton topology, which is a tree structure of bones typically fixed for a character, and the local joint rotations of the bones, which determine the actual pose of the skeleton. Forward kinematics (fk) is used to express the pose in 3D space, while inverse kinematics (ik) recovers the local joint rotations given the 3D points and skeleton. Importantly, the 3D pose is determined by both the local joint rotations and the individual bones, which are 3D vectors. This is particularly crucial for bilateral structures, such as humans, as directionality can be expressed either through the direction of the bone or a joint rotation. Many existing human skeleton topologies use a mix of both methods. To visualize this, we show three reset poses, where all local rotations are set to the unit rotation, in Figure 2 (d)–(f).

This structural ambiguity makes learning a generative model for multiple skeleton topologies challenging, especially considering that many motion models focus solely on local pose parameters and rarely account for the skeleton topology, which defines the local rotation space.

One approach to address this challenge is to retarget various skeletons to a unified representation, as seen in AMASS (Mahmood et al., 2019) and HumanML3D (Guo et al., 2022). While this solves the

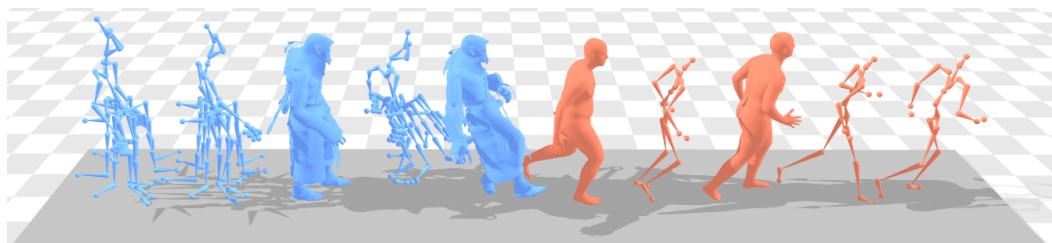

Figure 1: Industry animation projects rely on custom articulated human skeletons while generative models for motion generation require a unified representation, often SMPL. In this work, we learn a unified human representation which allows for easy translation between various skeletons. In contrast to other skeleton-agnostic embeddings, our approach learns to align different skeletons in Euclidean space, greatly simplifying the learning process. We provide visualizations of both the underlying articulated skeletons and the skinned meshes, which represent the final output.

problem of data scarcity and structural ambiguity, the process is complex, error-prone[1], and often undesired by artists, who prefer to work directly in the topology of the target character.

Learning topology-agnostic representations in configuration space has seen progress in recent years. Aberman et al. (2020) introduced a learned representation and operations for retargeting between different humanoid skeletons. Similarly, Skeleton-Agnostic Motion Embedding (SAME) (Lee et al., 2023) employs a graph-convolutional autoencoder to retarget motion snippets across skeletons. More recently, AnyTop (Gat et al., 2025) explicitly modeled skeletal structures using kinematic graphs and limb names, extending retargeting capabilities to non-humanoid skeletons.

Despite these advancements, topology-agnostic methods still face limitations. To handle topologies with different reset poses, they must learn poses relative to a common neutral pose (often a T-pose), which may not be perfectly aligned across skeletons. Additionally, bilateral structures must remain consistent across all skeletons, requiring rotations and bone directions to uniformly determine handedness - a condition rarely met, as illustrated in Figure 2 (d)–(f). Consequently, current methods must first align skeleton topologies for inverse kinematics consistency, often relying on proprietary software such as MotionBuilder (Autodesk, 2025). This alignment process is labor-intensive and error-prone, making it difficult to extend to new topologies.

In this work, we make a simple yet effective observation: while skeletons can have vastly different configuration spaces - defined by complex, non-linear systems with multiple solutions - they are inherently aligned in Euclidean space by construction. In realized 3D space, points are quasi-independent of their hierarchical structure, greatly simplifying the learning process. Reconstruction loss in Euclidean space is straightforward, whereas hierarchical structures require careful weighting due to the influence of early elements in the kinematic chain.

One challenge of using realized 3D joints is the entanglement of pose and human shape, such as varying bone lengths within the same topology. To address this, we introduce a learning scheme to explicitly disentangle body shape and human pose in our learned representation. Finally, to recover articulated poses, we learn inverse kinematics. Our experiments show that slightly more than five minutes of motion capture data is sufficient to learn consistent ik for a human skeleton topology, which indicates that our framework enables us to add new topologies easily.

In summary, our contributions are three-fold: 1. We introduce a learned latent representation that leverages human poses in Euclidean space rather than their various configuration spaces. 2. We provide a simple yet effective learning strategy to disentangle pose and body shape for the learned latent representation. 3. We demonstrate that our topology-agnostic representation can be utilized for three downstream tasks: motion retargeting, text-to-motion generation, and motion captioning.

## 2 RELATED WORK

**Motion Latent Space**: Early deep-learning-based motion representations (Holden et al., 2015; Villegas et al., 2018) utilized convolutional neural networks to enable tasks such as denoising, interpo-

---

[1]i.e. `https://github.com/EricGuo5513/HumanML3D/issues/119`

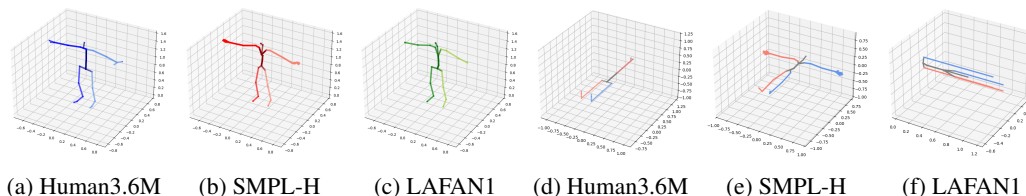

(a) Human3.6M    (b) SMPL-H    (c) LAFAN1    (d) Human3.6M    (e) SMPL-H    (f) LAFAN1

Figure 2: Heterogeneity of human motion datasets: Human motion datasets exhibit significant variability in skeletal topologies. (a) - (c) To illustrate this, we retarget a given pose to three distinct human topologies. Human3.6M (2a) represents only the thumb and index finger, while SMPL-H (2b) models a full five-finger hand. In contrast, LAFAN1 (2c) omits fingers entirely. (d) - (f) Neutral pose for various skeleton topologies: The neutral pose is depicted for different skeleton topologies, where all joint rotations are reset to $\theta = \mathbf{0}$. Note that the neutral pose (reset pose) does not necessarily represent a realistic human pose. Human3.6M (2d): This topology uses distinct bones with opposite directions for the hips, while the shoulders share the same bone direction. SMPL-H (2e): This topology expresses a T-pose, with all bilateral limbs pointing in different directions. LAFAN1 (2f): This topology has the hip and collarbone bones oriented in different directions, while all other bones point in the same direction. (Larger plots in Figure 9 and Figure 10)

lation or similarity search. Aberman et al. introduced a joint space for transfering 2D motion to a 3D skeleton. Deep-Phase networks (Starke et al., 2022) are Markov-chain style neural networks with a periodic *heartbeat* to effectively model temporal data. PMNet (Lim et al., 2019) learns pose and motion separately for retargeting, to better adapt to various character sizes. Aberman et al. (2020) introduce a skeleton-agnostic representation which utilizes a minimal *primal* skeleton for motion modelling. Skeleton-Agnostic Motion Embedding (Lee et al., 2023) (SAME) learns an embedding space over a large set of augmentations of an initial set of skeletons, learning to disentangle different topologies. CAR (Cao & Yang, 2024) introduces a purely algorithmic solution for retargeting between different skeletons. MoMa (Martinelli et al., 2024) proposes a masked auto-encoder to retarget motion sequences between different skeletons. AnyTop Gat et al. (2025) explicitly encodes the graph topology as well as node names into the model, allowing for a joint learning of completely different topologies. Motion2Motion (Chen et al., 2025) allows transferring motion across different topologies, utilizing only a few corresponding samples.

SMPL (Loper et al., 2015) is a widely-used 3D human body model that represents the human body as a mesh with a fixed topology. It is designed to capture a wide range of human shapes and poses using a small number of parameters. SMPL-H (Romero et al., 2017) extends SMPL with articulated hands.

**Generative Motion Generation**: Early generative models for motion synthesis include the works of Komura et al. (2017), Ghorbani et al. (2020), and Ling et al. (2020). More recently, text-to-motion Guo et al. (2022) models have seen increased popularity, ranging from data-space based methods (Tevet et al., 2023; Li et al., 2026) to masked prediction models (Guo et al., 2024) to latent diffusion models (Zhang et al., 2024; Meng et al., 2025; Uchida et al., 2025). All these models utilize a unified representation, usually HumanML3D Guo et al. (2020), which cannot be readily applied to different skeleton topologies. Furthermore, latent-space based methods such as MoMask (Guo et al., 2024) and MARDM (Meng et al., 2025) explicitly model motion at a fixed frame rate only, making industry adoption more challenging.

**Inverse Kinematics**: Inverse Kinematics is a non-linear, often non-unique, transformation from Euclidean space to joint configuration space. Analytical solutions exist for simple kinematic chains, but for more complex ones, such as the kinematic trees that make up humans, numerical methods are required. However, these methods often rely on a good initial estimate for convergence. Neural Inverse Kinematics (Bensadoun et al., 2022) utilizes a Gaussian Mixture Model to model the solution space. In contrast, Lu et al. (2022) utilizes a simple feed-forward neural network to provide a first estimate and then utilizes an iterative refinement to obtain the final configuration parameters. We follow their simple yet effective approach in utilizing a simple MLP to provide a first estimate, and, if needed, apply gradient-descent based optimization to align the end-effectors.

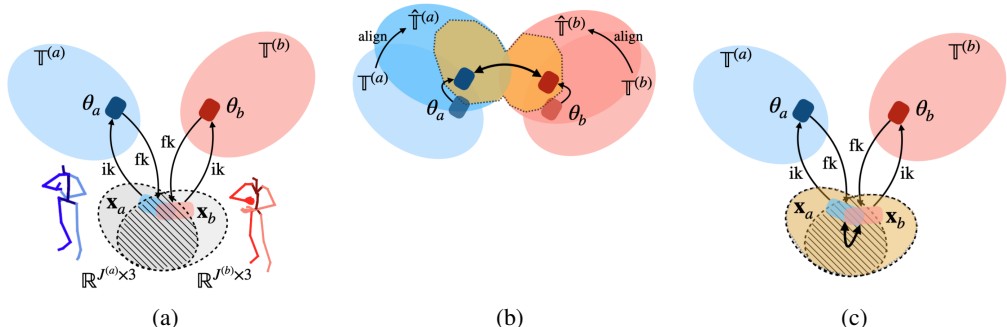

(a)                                    (b)                                    (c)

Figure 3: An articulated human pose $(\theta, o, \mathcal{S}) \in \mathbb{T}$ can be realized in Euclidean space via forward kinematics (fk). Conversely, joint rotations can be recovered from the 3D Euclidean points $\mathbf{x} \in \mathbb{R}^{J \times 3}$ using inverse kinematics (ik). State-of-the-art skeleton-agnostic methods first align their configuration spaces to learn a joint representation in configuration space (3b). In contrast, we directly learn to align different poses in Euclidean space, which is inherently aligned by construction (3c). We then learn the ik back to configuration space, ensuring that our learned representations are fully compatible with the original configuration spaces.

## 3 METHOD

### 3.1 PRELIMINARIES

An articulated human pose with $J$ joints is represented by a set of local joint rotations $\theta \in \mathbb{SO}(3)^J$, bone offsets $o \in \mathbb{R}^{J \times 3}$, and a skeleton $\mathcal{S}$, which defines the kinematic chains as a tree structure for the given topology. The articulated pose triplet $(\theta, o, \mathcal{S}) \in \mathbb{T}$ is an element of the configuration space $\mathbb{T}$.

In this work, we adopt the 3D Cartesian coordinates to represent human poses in Euclidean space, which is a common practice. To compute the 3D Cartesian coordinates of the human pose, $\mathbf{x} \in \mathbb{R}^{J \times 3}$, forward kinematics (fk) is applied:

$$\mathbf{x} = \text{fk}(\theta, o, \mathcal{S}) \tag{1}$$

Here, the position of each joint in $\mathbf{x}$ is determined by both the joint rotations $\theta$ and the bone offsets $o$. Importantly, there can be infinitely many combinations of rotations and offsets that produce the same $\mathbf{x}$.

To recover the joint rotations from the Cartesian coordinates, inverse kinematics (ik) is used:

$$\hat{\theta} = \text{ik}(\mathbf{x}, o, \mathcal{S}) \tag{2}$$

where $\hat{\theta}$ represents one possible solution. Unlike forward kinematics, which is linear, inverse kinematics is non-linear and may have multiple solutions. While analytical solutions exist for simple kinematic chains, more complex structures, such as the kinematic trees that define human skeletons, require numerical methods. In this work, we approximate the inverse kinematics function ik using a neural network.

### 3.2 METHOD OVERVIEW

Our model consists of three main components: (1) the translation-to-anchor module, (2) the disentangled anchor representation autoencoder, and (3) the neural inverse kinematics module. While the entire architecture is trained end-to-end, each sub-component can also be trained independently, which is particularly useful when integrating a new topology into a pretrained model without retraining the other components. This enables us to leverage knowledge of the model pretrained on a larger scale of data. A comprehensive overview of our model is presented in Figure 4. All models are trained using a reconstruction loss, specifically the Mean Squared Error (MSE).

**Anchor Representation**: Similar to other retargeting methods such as Aberman et al. (2020), we utilize a common representation to learn human poses. However, unlike methods that operate in

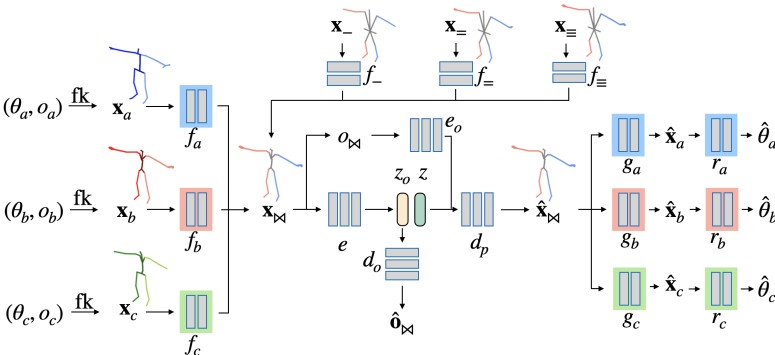

Figure 4: Model overview: Given articulated pose $(\theta, o)$ for a given topology, we learn a topology-dependent projection $f$ that maps the Cartesian coordinates $\mathbf{x}$ to the anchor representation $\mathbf{x}_{\bowtie}$. Alternatively, we provide generic learned mappings $f_-$, $f_=$, and $f_{\equiv}$ which map generic reduced skeletons to the anchor. Given the anchor $\mathbf{x}_{\bowtie}$ we use encoder $e$ to produce pose and shape embeddings $z_o$ and $z$. We utilize shape decoder $d_o$ to regress the input shape parameter $o_{\bowtie}$ and pose decoder $d_p$ to regress the anchor. Note that pose decoder takes as input both $z$ and a learned projection of the input shape $o_{\bowtie}$. We regress per-topology 3D representations $\mathbf{x}$ with $g$ and learn inverse kinematics function $r$ to obtain articulated pose parameters $\theta$.

configuration space and require a minimal sub-skeleton, our approach employs a flexible common representation, referred to as the *anchor representation*, which can handle arbitrary complexity and even act as a superset of many skeletons. This flexibility is achieved by learning the representation directly from poses in Euclidean space rather than configuration space.

We use the 3D joints of SMPL-H as our anchor topology due to its widespread availability and ability to represent complex poses, including detailed finger movements (see Figure 9b). The anchor pose is represented as $\mathbf{x}_{\bowtie} \in \mathbb{R}^{J_{\bowtie} \times 3}$ in Euclidean space, distinguishing our method from prior works.

To convert poses to the anchor representation, we employ two approaches:

1. *Paired Data*: For select datasets, we use off-the-shelf retargeting software to carefully retarget the dataset to SMPL-H. This provides paired data for training the pose-to-anchor function $f$ and the anchor-to-pose function $g$.

2. *Preset Representations*: For datasets without paired SMPL-H data, we define three simplified pose presets: $\mathbf{x}_-$ (minimal skeleton without fingers), $\mathbf{x}_=$ (skeleton with two fingers: thumb and index), and $\mathbf{x}_{\equiv}$ (skeleton with all five fingers). Regular human pose skeletons can be converted to one of these presets by selecting the appropriate joints. We pre-train $f_-$, $f_=$, and $f_{\equiv}$ to translate from these presets to the anchor pose, leaving only the anchor-to-pose function $g$ to be learned for new datasets.

**Pose Representation Autoencoder**: One downside of utilizing the 3D Euclidean pose representation is its entanglement between pose and body shape, which in our work are represented by the skeleton offset vectors $o \in \mathbb{R}^{J \times 3}$. However, we want our learned pose representation to be independent of the body shape. To facilitate this, we learn an autoencoder where the encoder $e$ predicts two latent vectors, $z_o$, and $z$, and where $z$ is our latent disentangled pose representation, which we utilize for downstream tasks. To recover the 3D Euclidean pose $\hat{\mathbf{x}}_{\bowtie}$, we learn decoder $d_p$, conditioned on the ground-truth human shape $o_{\bowtie}$. Note that $o_{\bowtie}$ can be directly regressed from the 3D points $\mathbf{x}_{\bowtie}$, assuming that the direction of the skeleton in reset pose is known.

To disentangle pose and body shape, we utilize two strategies: first, we learn auxiliary decoder $d_o$ to directly regress $\hat{o}_{\bowtie}$ ($\mathcal{L}_o$). Second, we train decoder $d_p$ with augmented $o_{\bowtie}$, where the skeleton is randomly scaled up, down and randomly sampled from another character. During training, the decoder then has to recover not just the original $\mathbf{x}_{\bowtie}$ but also the ones conditioned on the augmented offsets, while provided with the same $z$ ($\mathcal{L}_{\text{aug}}$).

**Training Objective**: Our autoencoder training objective is defined as follows:

$$\mathcal{L}_{\text{ae}} = \mathcal{L}_{\text{rec}} + \lambda_o \cdot \mathcal{L}_o + \lambda_{\text{aug}} \cdot \mathcal{L}_{\text{aug}} \tag{3}$$

where $\mathcal{L}_{\text{rec}}$ is the reconstruction loss of the original pose, $\mathcal{L}_{\text{aug}}$ is the reconstruction loss for shape augmentations, and $\mathcal{L}_o$ is the reconstruction loss for the offset.

Our end-to-end loss is defined as:

$$\mathcal{L} = \lambda_{\bowtie}\big(\mathcal{L}_{\to\bowtie} + \mathcal{L}_{\bowtie\to}\big) + \lambda_{\text{ik}} \cdot \mathcal{L}_{\text{ik}} + \mathcal{L}_{\text{ae}} \tag{4}$$

where $\mathcal{L}_{\to\bowtie}$ and $\mathcal{L}_{\bowtie\to}$ are the losses for converting to and from the anchor representation, respectively, and $\mathcal{L}_{\text{ik}}$ represents the inverse kinematics loss. We use mean squared error for all loss functions.

**Implementation Details**: Implementation details are described in Appendix A.3.

## 4 EXPERIMENTS

In Section 4.1, we evaluate the inverse kinematics, which are crucial to recover articulated poses for downstream tasks. Section 4.2 discusses the various design choices, while we discuss downstream applications such as retargeting, text-to-motion, and motion captioning in Section 4.3.

### 4.1 INVERSE KINEMATICS

Table 1: Inverse kinematics mean error (in cm) and accuracy for different datasets.

| Dataset | MPJPE | Acc@10 | Acc@5 | Acc@1 |
|---|---|---|---|---|
| AMASS (Mahmood et al., 2019) | 1.40 | 98.20% | 86.87% | 1.30% |
| LAFAN1 (Harvey et al., 2020) | 0.86 | 99.22% | 94.92% | 12.31% |
| Motorica (Valle-Pérez et al., 2021) | 1.24 | 99.08% | 88.48% | 0.18% |
| Human3.6M (Ionescu et al., 2013) | 0.94 | 98.11% | 94.06% | 11.25% |
| FineDance (Li et al., 2023) | 1.66 | 95.68% | 69.48% | 0.02% |

Table 2: Ablation of inverse kinematics functions. Mean error is measured in cm.

| | Human3.6M | | | | |
|---|---|---|---|---|---|
| Method | MPJPE | Acc@10 | Acc@5 | Acc@1 | #param |
| Linear | 1.96 | 92.21% | 61.97% | 0.00% | 18,624 |
| MLP-2 | 1.24 | 99.33% | 89.57% | 1.35% | 148,160 |
| MLP-5 | 0.93 | 98.41% | 95.24% | 15.18% | 936,128 |
| MLP-5 + Opt | 0.13 | 100.00% | 100.00% | 99.98% | 936,128 |

We show qualitative results of our inverse kinematics for five different datasets and skeleton topologies in Figure 11 and in Appendix A.2. We adopt the evaluation metrics established in prior work (Bensadoun et al. (2022)) to assess the performance of our learned inverse kinematics (IK) method, specifically mean distance and accuracy. However, we introduce a stricter definition of accuracy: rather than determining accuracy based on whether the mean joint error falls below a given threshold, we instead threshold the maximum limb mismatch. This adjustment ensures that a single poorly aligned joint is not overshadowed by the average of well-aligned joints. We evaluate accuracy at three thresholds: $10\,\text{cm}$, $5\,\text{cm}$, and $1\,\text{cm}$. In Table 1, we evaluate the capabilities of our learned inverse kinematics for five different datasets. We find that our model works best on Human3.6M and LAFAN1, which contain common motion sequences, such as walking, running or sitting, while dance datasets, such as Motorica and FineDance, are more difficult, due to them containing more extreme poses on average. AMASS contains a wide variety of motions and, crucially, of body shapes, which makes the problem more challenging.

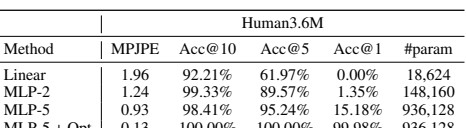
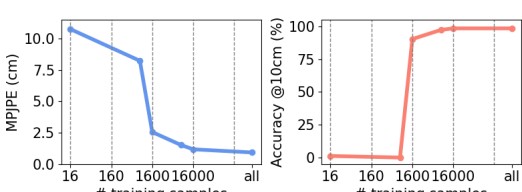

Figure 5: Evaluation of mean positional error (in cm) and accuracy of the learned inverse kinematics on the Human3.6M dataset, given varying numbers of training samples. The model demonstrates the ability to generalize to the test set with approximately 16,000 training samples, equivalent to around 5.33 minutes of data at a frequency of 50 Hz.

In Table 2, we ablate the inverse kinematics function. Surprisingly, even a linear function provides a reasonable approximation in many instances. This can also be visually inspected in Appendix A.2.

**Generalization Performance**: The primary objective of our representation is to enable various generative modeling capabilities across different human motion representations and even new topologies not encountered during the training of either the generative model or the pose representation. When using a new, unseen topology, it is necessary to learn a Euclidean space decoder $g(\cdot)$ and an inverse kinematics function $r(\cdot)$ specific to that topology. In Figure 5, we evaluate the number of training samples required for the learned inverse kinematics to generalize on the Human3.6M (Ionescu et al.,

2013) dataset. Our experiments indicate that the model can generalize to the unseen test set with just 5.33 minutes of training data.

## 4.2 RECONSTRUCTION

Table 3: Autoencoder evaluation of the reconstruction and disentanglement in $\mathrm{cm}$.

| $z$-size | $\mathcal{L}_o$ | $\mathcal{L}_{\mathrm{aug}}$ | reconstruction MPJPE ↓ | disentanglement MPJPE ↓ |
|---|---|---|---|---|
| 128 | ✓ | ✓ | 1.41 | 2.10 |
| 256 | | | 0.90 | 3.50 |
| 256 | | ✓ | 0.86 | 1.70 |
| 256 | ✓ | | 0.90 | 3.46 |
| 256 | ✓ | ✓ | **0.78** | **1.26** |
| 512 | ✓ | ✓ | 0.66 | 1.22 |

Table 4: Retargeting evaluation on the Mixamo evaluation protocol established by Aberman et al. (2020).

| Method | Intra ↓ | Cross ↓ |
|---|---|---|
| Copy rotations | 8.86 | N/A |
| NKN (Villegas et al., 2018) | 6.24 | 243 |
| PMnet (Lim et al., 2019) | 5.72 | N/A |
| CycleGAN adaptation (Aberman et al., 2020) | 7.66 | 8.97 |
| Aberman et al. (2020) (no $L_{\mathrm{adv}}$) | **0.47** | 3.81 |
| Aberman et al. (2020) (full approach) | 2.76 | 2.25 |
| SAME (Lee et al., 2023) | 2.91 | 2.47 |
| Ours (same latent) | 5.31 | **1.62** |

In Table 3, we qualitatively evaluate our method and latent space design choices, particularly focusing on how they affect the disentanglement of pose and body shape. We use two metrics for evaluation: reconstruction, which measures the Euclidean distance between the input pose and the reconstructed pose in centimeters, and disentanglement, which assesses how well the latent decoder generates the same pose under different body shapes, also measured in centimeters.

For the experiments, we utilize a withheld test set from AMASS and cluster the poses into 128 pose clusters and select 64 random poses from each cluster to ensure that more common poses do not overshadow extreme and rare poses. Similarly, we cluster the offsets into 32 clusters and use the cluster centers as our input body shapes. This approach ensures a wide physical variety without the risk of results being dominated by average-height individuals, which are more prevalent in many datasets. In Appendix Figure 13, we show four random pose samples from four different clusters with two different offsets.

Disentanglement measures how well the decoder is conditioned on the provided offsets $o_{\bowtie}$ and how effectively the latent space $z$ represents the same pose under different body shape offsets. We measure this by first encoding a pose $\mathbf{x}_{\bowtie}$ into the latent space $z$ and then decoding the pose under the 32 selected offsets, which represent different human body shapes, such as very small, medium, or tall individuals. The disentanglement metric represents the average distance error across these predicted results.

As expected, increasing the latent space dimension improves reconstruction. However, for all our experiments, we set the size of the latent space to 256 to remain comparable with the data representation sizes used in generative human motion modeling. We find that $\mathcal{L}_{\mathrm{aug}}$ is crucial for disentanglement.

We qualitatively discuss the disentanglement of the latent space in Appendix Section A.5.

## 4.3 DOWNSTREAM APPLICATIONS

### 4.3.1 RETARGETING

Retargeting involves transferring motion from one skeleton to another, and our work is closely related to other efforts in this space, such as Aberman et al. (2020) and SAME (Lee et al., 2023). For this experiment, we follow the established retargeting benchmark on the Mixamo dataset, as outlined in Aberman et al. (2020), and present our results in Table 4. Qualitative results are shown in Figure 6.

To demonstrate the extensibility of our method to new topologies, we learn our embedding space using only the AMASS dataset. For learning the different characters required for retargeting and inverse kinematics, we use approximately 10 minutes of training data - significantly fewer training samples than those used in Aberman et al. (2020) or SAME. This data is used to train the from-anchor model $g$ and the ik model $r$.

To avoid the need to retarget SMPL-H to Mixamo, we utilize the reduced preset $\mathbf{x}_{\equiv}$.

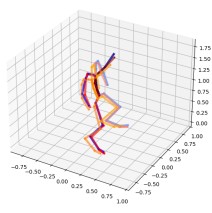 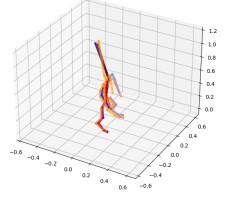 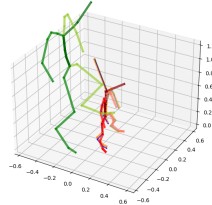

(a) *Goblin → Goblin* (cross)    (b) *Mousey → Mousey* (cross)    (c) *Goblin → Mousey* (intra)

Figure 6: Retargeting results on the Mixamo dataset, comparing our and state-of-the-art (Aberman et al., 2020) results against ground-truth. (a) and (b) show results on the cross-structure retargeting task for characters *Goblin* and *Mousey*. (c) shows intra-structural retargeting from *Goblin* to *Mousey*. Note that our representation maintains the pose of the source pose (leaning forward) instead of adjusting it to the target character.

Table 5: Text-to-motion evaluation on HumanML3D (Guo et al., 2022): We assess the performance of our representation across two distinct text-to-motion models: the data-space-based method MDM and the latent diffusion model MARDM.

| Method | R Precision↑ | | | FID ↓ | MM Dist.↓ | Diversity→ | MModality ↑ |
|---|---|---|---|---|---|---|---|
| | Top 1 | Top 2 | Top 3 | | | | |
| MDM (Tevet et al., 2023) | $0.320^{\pm.005}$ | $0.498^{\pm.005}$ | $0.611^{\pm.007}$ | $\mathbf{0.544}^{\pm.007}$ | $5.566^{\pm.027}$ | $9.559^{\pm.086}$ | $2.799^{\pm.072}$ |
| MDM (ours) | $\mathbf{0.419}^{\pm.001}$ | $\mathbf{0.631}^{\pm.002}$ | $\mathbf{0.742}^{\pm.002}$ | $0.663^{\pm.008}$ | $\mathbf{3.540}^{\pm.001}$ | $9.950^{\pm.077}$ | $2.174^{\pm.019}$ |
| MARDM-DDPM (Meng et al., 2025) | $\mathbf{0.492}^{\pm.007}$ | $0.690^{\pm.005}$ | $0.790^{\pm.005}$ | $\mathbf{0.116}^{\pm.004}$ | $3.349^{\pm.010}$ | $10.613^{\pm.105}$ | $2.470^{\pm.053}$ |
| MARDM-DDPM (Ours) | $\mathbf{0.492}^{\pm.000}$ | $\mathbf{0.694}^{\pm.000}$ | $\mathbf{0.794}^{\pm.000}$ | $0.238^{\pm.000}$ | $\mathbf{3.236}^{\pm.000}$ | $10.897^{\pm.170}$ | $2.234^{\pm.089}$ |

**Intra-Structural Retargeting** tests how well a motion representation allows for the transfer of a pose of a given topology $\mathbb{T}$ and proportions $o_a$ to another skeleton with the same topology but different proportions $o_b$. Directly translating from one topology to another but utilizing the same encoding $z$ will result in the same pose, which might not be desired for character transfer, as can be seen in Figure 6c, where the Goblin character leans forward while the *Mousey* character of the same pose leans backwards. In this sense, our method is similar to *Copy Rotations*, which copies the exact pose. Our method, however, produces better results than Copy Rotations as it is explicitly conditioned on $o$, and thus can make some adjustments to the pose to better align joints.

**Cross-Structural Retargeting** tests how well a motion representation allows for the transfer of a pose in a given topology $\mathbb{T}_1$ to a new topology $\mathbb{T}_2$. To obtain those two different topologies, extra bones are added in-between existing ones in the neck, legs and arms, by *splitting* the original bone. This task is extremely difficult when learning in a hierarchical rotation space but becomes trivial when using Euclidean space, as the *salient* joints remain at their original locations.

### 4.3.2 TEXT-TO-MOTION

To evaluate the suitability of our representation for the text-to-motion task, we follow the established evaluation protocols for the HumanML3D benchmark (Guo et al., 2022). We test our approach with two generative models, the data-space-based method MDM (Tevet et al., 2023), and the latent diffusion model MARDM (Meng et al., 2025). As shown in Table 5, our representation performs competitively compared to the original HumanML3D representation.

Our representation offers several advantages over HumanML3D. Although HumanML3D relies on 3D keypoints and velocities that encode both human shape and framerate, our representation is inherently more flexible. It can directly represent articulated motion, including more complex motions such as hand movements. Furthermore, our representation enables seamless translation to different skeletons, as demonstrated in Figure 7.

We discuss the slightly higher FID scores in Appendix A.4.

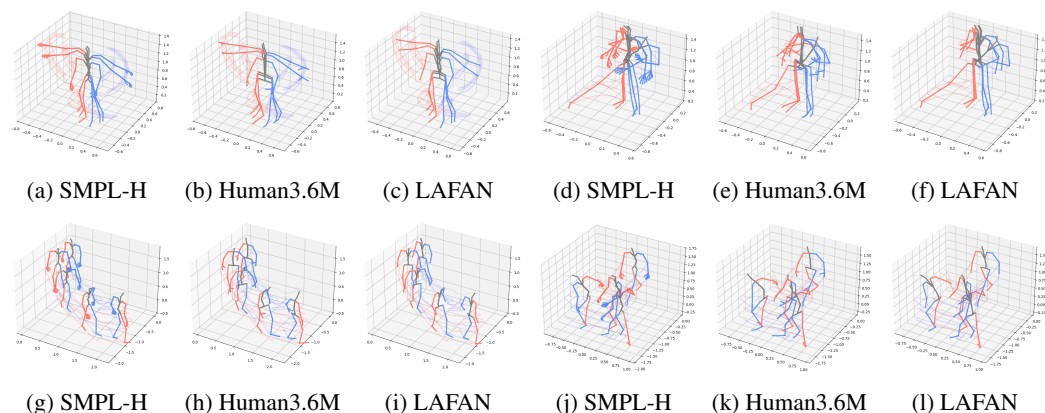

(a) SMPL-H   (b) Human3.6M   (c) LAFAN   (d) SMPL-H   (e) Human3.6M   (f) LAFAN

(g) SMPL-H   (h) Human3.6M   (i) LAFAN   (j) SMPL-H   (k) Human3.6M   (l) LAFAN

Figure 7: Generative results of MDM in human topologies SMPL-H, Human3.6M and LAFAN. Note that that the same latent sequence was used to extract the motion in the representations. Captions: (a) - (c): *A person is doing jumping jacks.*; (d) - (f): *A person puts their hands together and kicks their right foot.*; (g) - (i): *A person walking side to side.*; (j) - (l): *A person walks forward with right foot, then trips and continues walking forward.* Better viewed in the supplementary video.

Table 6: Text generation and motion-text-alignment scores on HumanML3D.

| Method | Text Quality | | | R Precision↑ | | | |
|---|---|---|---|---|---|---|---|
| | Bleu@1↑ | Rouge↑ | Bert Score↑ | Top 1 | Top 2 | Top 3 | MM-Dist↓ |
| MotionGPT (Jiang et al., 2023) | $0.291^{\pm0.001}$ | $0.282^{\pm002}$ | $0.221^{\pm0.002}$ | $0.435^{\pm0.004}$ | $\underline{0.627}^{\pm0.004}$ | $0.730^{\pm0.001}$ | $\underline{3.581}^{\pm0.008}$ |
| MotionAgent (Wu et al., 2025) | $\mathbf{0.325}^{\pm0.003}$ | $\mathbf{0.339}^{\pm001}$ | $\mathbf{0.290}^{\pm0.001}$ | $\mathbf{0.499}^{\pm0.004}$ | $\mathbf{0.698}^{\pm0.005}$ | $\mathbf{0.801}^{\pm0.004}$ | $\mathbf{3.243}^{\pm0.009}$ |
| Ours (#clusters 512, single) | $0.311^{\pm0.001}$ | $0.306^{\pm001}$ | $0.244^{\pm0.002}$ | $0.414^{\pm0.004}$ | $0.598^{\pm0.005}$ | $0.703^{\pm0.003}$ | $3.761^{\pm0.014}$ |
| Ours (#clusters 256, 4-set) | $\underline{0.316}^{\pm0.001}$ | $\underline{0.314}^{\pm002}$ | $0.246^{\pm0.001}$ | $0.436^{\pm0.005}$ | $0.618^{\pm0.003}$ | $0.722^{\pm0.003}$ | $3.729^{\pm0.018}$ |
| Ours (#clusters 512, 4-set) | $0.302^{\pm0.003}$ | $0.305^{\pm0.002}$ | $0.237^{\pm0.002}$ | $0.419^{\pm0.003}$ | $0.602^{\pm0.004}$ | $0.706^{\pm0.007}$ | $0.378^{\pm0.015}$ |
| Ours (#clusters 1024, 4-set) | $0.306^{\pm0.001}$ | $0.308^{\pm0.001}$ | $\underline{0.246}^{\pm0.001}$ | $\underline{0.439}^{\pm0.004}$ | $0.623^{\pm0.002}$ | $\underline{0.731}^{\pm0.002}$ | $0.365^{\pm0.008}$ |

### 4.3.3 CAPTIONING

Another relevant generative model task is motion captioning, where a large language model generates a caption for a provided input motion. We compare our method to two existing approaches, MotionGPT (Jiang et al., 2023) and MotionAgent (Wu et al., 2025), and retrain MotionAgent on two variants of our representation: (1) $k$-means clustering of our latent space, and (2) $k$-means clustering of groups of four adjacent frames in latent space to encode motion, similar to the VQ-VAE utilized in both MotionGPT and MotionAgent. Our results are presented in Table 6, with qualitative results shown in Appendix A.8.

Our experiments reveal that even the simple discretization method $k$-means can be effectively utilized for motion captioning. We attribute the slightly lower performance of our representation to the absence of global transformation, which makes distinguishing relative motion more challenging. This is further supported by the slightly larger multi-modal distance observed in our results.

## 5 CONCLUSION

We presented a simple yet effective approach to human pose representation that leverages Euclidean space, eliminating the need for alignment in configuration space. This simplicity allows our method to flexibly handle diverse articulated skeleton topologies. By introducing a learning strategy to disentangle pose and body shape, we ensure robustness and generalizability. Furthermore, we show that we learn inverse kinematics with minimal data requirements.

Our topology-agnostic representation demonstrates effectiveness across three downstream tasks: motion retargeting, text-to-motion generation, and motion captioning. The simplicity of our approach makes it scalable and practical.

ETHICS STATEMENT

**Societal Impact**: This method is particularly valuable to industry partners who work with diverse skeleton topologies and may face legal restrictions on using motion capture sequences. By reducing the labor and cost associated with manual retargeting, our method offers significant efficiency gains.

Moreover, by integrating modern generative technologies into artists' workflows, this method has the potential to enhance productivity. It frees artists from repetitive and labor-intensive tasks, allowing them to focus more on creative endeavors. This not only improves their efficiency but also fosters greater innovation and creativity in their work.

However, the automation of manual processes may impact certain jobs, particularly entry-level positions in animation and motion capture.

**LLM Usage**: We utilize LLMs for academic proofreading. We also used them for coding assistance, including algorithm implementation and visualization of experimental results. However, all research ideas were developed solely by the authors.

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

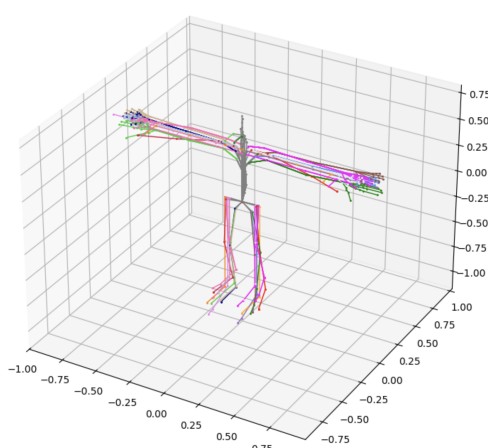

Figure 8: Various T-poses of different human motion datasets.

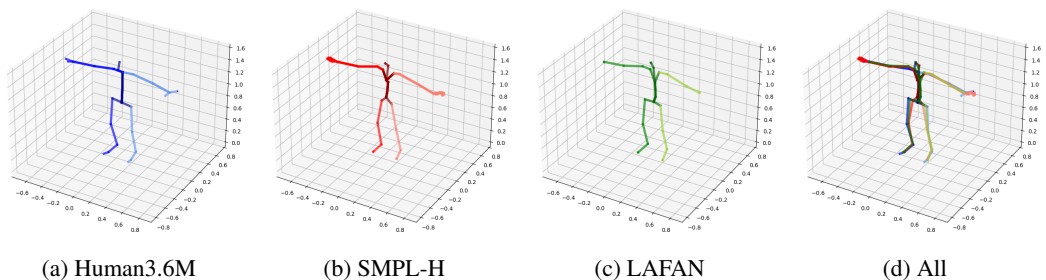

(a) Human3.6M      (b) SMPL-H      (c) LAFAN      (d) All

Figure 9: Heterogeneity of human motion datasets: Human motion datasets exhibit significant variability in skeletal topologies, which are influenced by the tools and use cases for which they are designed. To illustrate this, we retarget a given pose to three distinct human topologies. For instance, Human3.6M (9a) represents only the thumb and index finger, while SMPL-H (9b) models a full five-finger hand. In contrast, LAFAN (9c) omits fingers entirely. Overlaying all three skeletons (9d) reveals additional differences, such as variations in how the hip, neck, and spine are modeled across topologies.

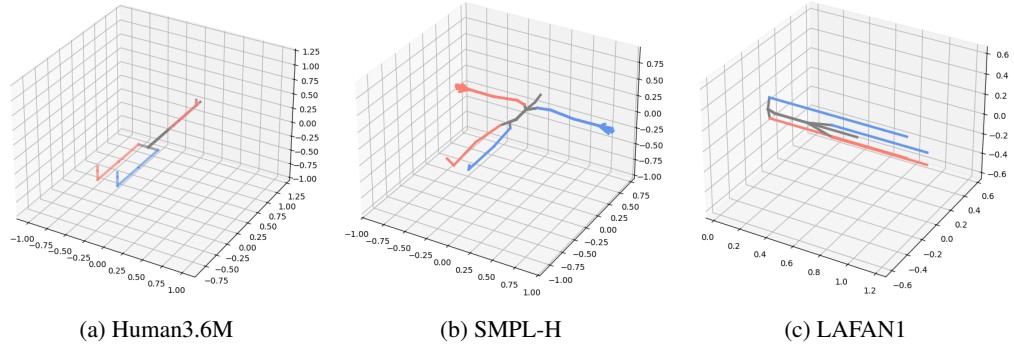

(a) Human3.6M      (b) SMPL-H      (c) LAFAN1

Figure 10: Reset pose for various skeleton topologies: The reset pose is depicted for different skeleton topologies, where all joint rotations are reset to $\theta = 0$. Note that the reset pose does not necessarily represent a realistic human pose. Human3.6M (10a): This topology uses distinct bones with opposite directions for the hips, while the shoulders share the same bone direction. SMPL-H(10b): This topology expresses a T-pose, with all bilateral limbs pointing in different directions. LAFAN1 (10c): This topology has the hip and collarbone bones oriented in different directions, while all other bones point in the same direction.

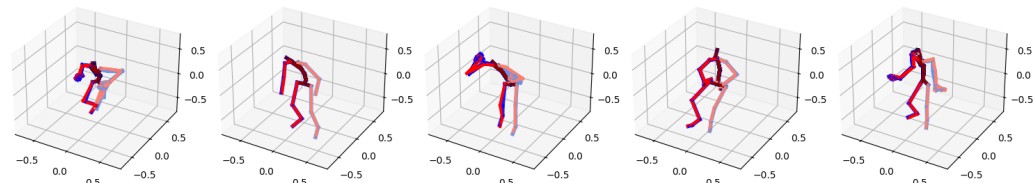

Figure 11: Results of the learned inverse kinematics on the unseen test sets for AMASS, LAFAN1, Motorica, Human3.6M and FineDance. The blue skeletons represent the ground truth 3D projections, while the red skeletons correspond to the predictions obtained by applying forward kinematics to the articulated pose parameters.

## A  APPENDIX

### A.1  DIFFERENCES IN STRUCTURE BETWEEN DATASETS AND TOPOLOGY

A T-pose is a default posing for many human skeleton topologies and is often used to align an animation rig with a 3D mesh. However, various datasets and human topologies might define a T-pose slightly differently, i.e., with different angles at the shoulders or at the legs, as can be seen in Figure 8.

Furthermore, human motion datasets exhibit significant variability in skeletal topologies, which are influenced by the tools and use cases for which they are designed. To illustrate this, we retarget a given pose to three distinct human topologies. For instance, Human3.6M (9a) represents only the thumb and index finger, while SMPL-H (9b) models a full five-finger hand. In contrast, LAFAN (9c) omits fingers entirely. Overlaying all three skeletons (9d) reveals additional differences, such as variations in how the hip, neck, and spine are modeled across topologies.

The reset pose of an articulated human skeleton is the pose when all local rotations are set to the unit rotation. We visualize different skeleton topologies under the reset pose in Figure 10. Note that the reset pose does not necessarily represent a realistic human pose. Human3.6M (10a): This topology uses distinct bones with opposite directions for the hips, while the shoulders share the same bone direction. SMPL-H (10b): This topology expresses a T-pose, with all bilateral limbs pointing in different directions. LAFAN1 (10c): This topology has the hip and collarbone bones oriented in different directions, while all other bones point in the same direction.

### A.2  INVERSE KINEMATICS

In Section 4.1, we discuss the performance of our inverse kinematics model. We test three model variants: a linear model that directly maps normalized 3D joint positions to 6d rotations, an MLP with two layers, and an MLP with five layers. For completeness, we also evaluate a post-processing gradient descent optimization procedure to obtain results as close as possible to the ground truth target points.

In Figure 11, we show inverse kinematics results of test poses on five different datasets with different human skeleton topologies, while in Figure 12, we show inverse kinematics results for various challenging poses on Human3.6M, comparing the linear model, our model, and our model with the post-processing optimization.

### A.3  IMPLEMENTATION DETAILS

For simplicity, we utilize simple MLPs as function approximators in all our learned functions, as our main focus is on the representation. For the inverse kinematics function $r$, we follow Lu et al. (2022) and also represent it as a 5-layer MLP. Our anchor encoder-decoder module has $2.8M$ parameters while the per-topology-representations have around $0.7M$ parameters. We train this model with the Adam optimizer and a learning rate of $0.0001$.

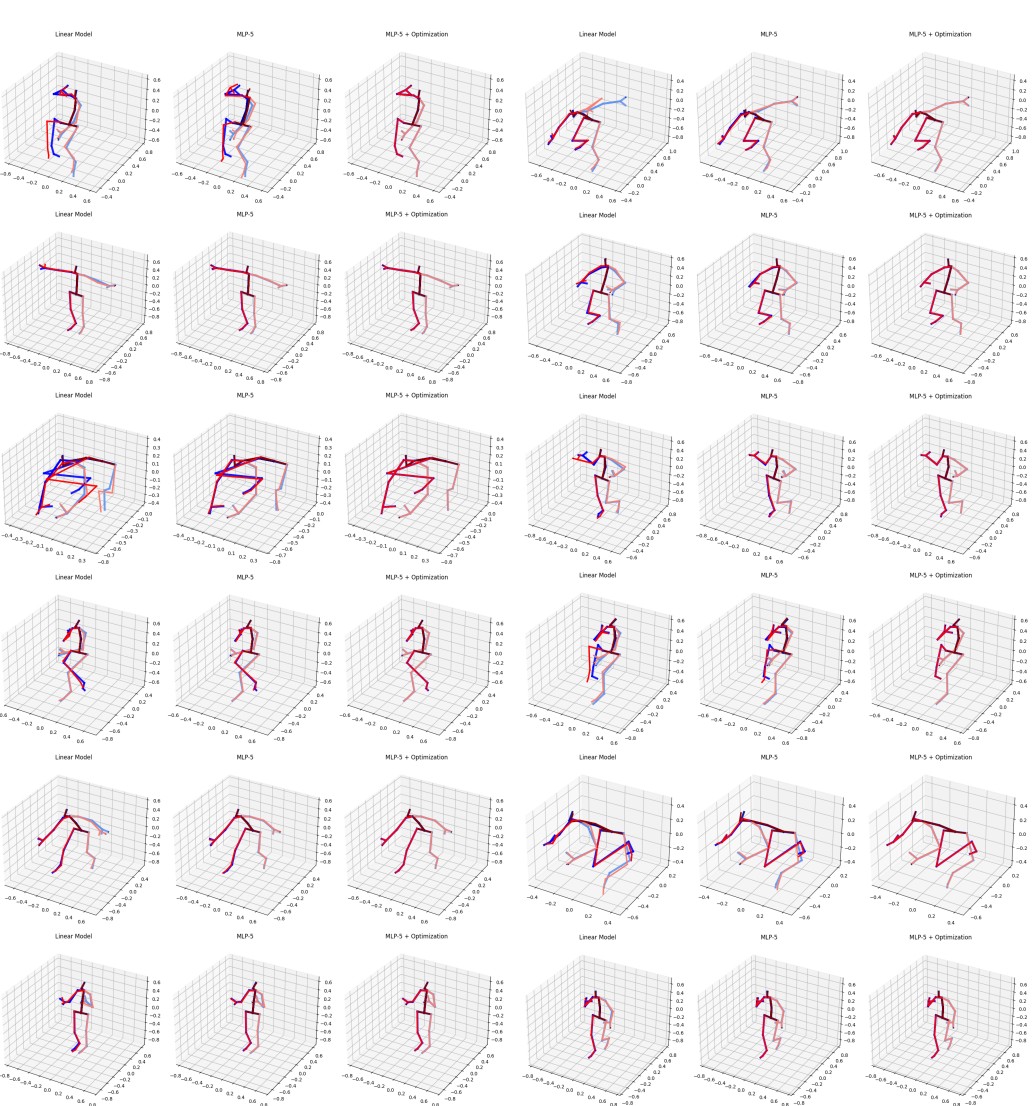

Figure 12: Results of the learned inverse kinematics on the unseen Human3.6M test set are presented. The blue skeletons represent the ground truth 3D projections, while the red skeletons correspond to the predictions obtained by applying forward kinematics to the articulated pose parameters. We compare the outputs of a linear model, our 5-layer MLP, and gradient descent-based post-optimization. The first row highlights two failure cases with the highest errors across the test set. Notably, even the linear model produces reasonable predictions in many challenging examples. The direct output of the MLP closely matches the ground truth, and the post-optimization step further refines the alignment, achieving the most accurate results.

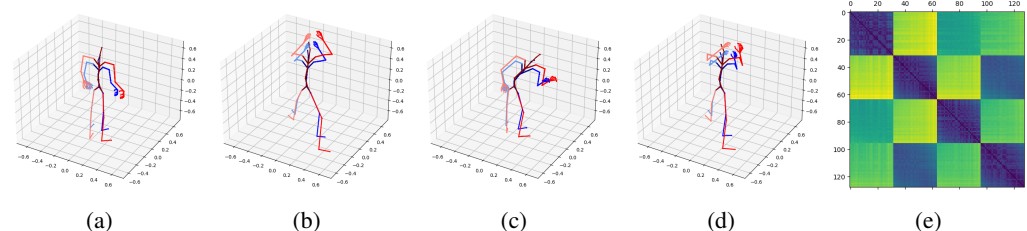

(a)   (b)   (c)   (d)   (e)

Figure 13: Visualization of shape augmentation for pose-shape disentanglement. In 13a, 13b, 13c and 13d, we show two shape variants of the same pose, while in 13e, we show the Euclidean distance in latent space between all four poses and their 32 variants, which are used for the evaluation.

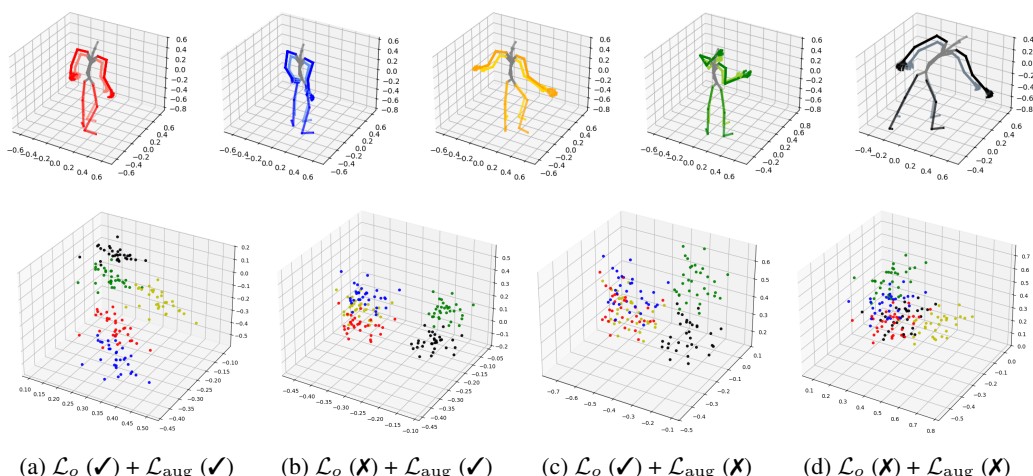

(a) $\mathcal{L}_o$ (✔) + $\mathcal{L}_{\mathrm{aug}}$ (✔) (b) $\mathcal{L}_o$ (✘) + $\mathcal{L}_{\mathrm{aug}}$ (✔) (c) $\mathcal{L}_o$ (✔) + $\mathcal{L}_{\mathrm{aug}}$ (✘) (d) $\mathcal{L}_o$ (✘) + $\mathcal{L}_{\mathrm{aug}}$ (✘)

Figure 14: PCA Visualization of Pose Variations with Different Offsets: We visualize five poses, each with 32 different offsets, as used in our experiments. The top row displays two instances of each pose, color-coded, with two different offsets. The bottom row shows the projection of the latent space $z$. By inspection, we observe that adding the augmentation loss $\mathcal{L}_{\mathrm{aug}}$ helps to separate different clusters of the same pose, while the shape loss $\mathcal{L}_o$ further pushes them apart in the latent space.

### A.4 DISCUSSION ON FID SCORE ON THE TEXT-TO-MOTION TASK

Our experiments reveal a slightly higher FID score for models trained on our representation, which we attribute to two main factors: distribution shift and noise accumulation during representation conversion. The distribution shift arises from converting our representation to the original HumanML3D format for evaluation, a challenge also observed in prior works such as STMC (Petrovich et al., 2024) and GENMO (Li et al., 2026), which directly utilize SMPL rather than the HumanML3D representation. Additionally, as in STMC, we only use the AMASS portion of HumanML3D and exclude HumanAct12 (Guo et al., 2020), as the pose formats between these datasets are not readily aligned[2].

### A.5 DISENTANGLEMENT EVALUATION

In the experiment (Section 4.2), we evaluate how disentangled the output poses of our pipeline are.

Here, we verify qualitatively the disentanglement of the latent space. Figure 13 illustrates the effect of shape augmentation on the latent space. As described in Section 4.2, we generate 32 random shape variants for each pose. In Figure 13e, we visualize the Euclidean distances between these 32 variants for four random test poses. As expected, the distances within a single pose, regardless of shape variations, are shorter than the distances across different poses.

---

[2]Aligning them is a pre-processing step in generating HumanML3D.

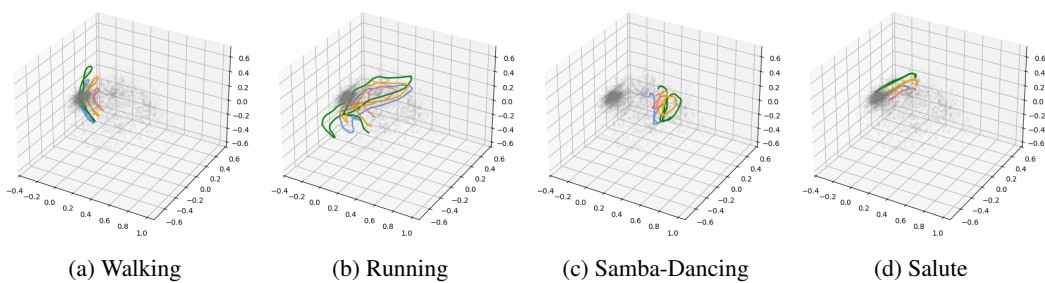

(a) Walking  (b) Running  (c) Samba-Dancing  (d) Salute

Figure 15: PCA-projection of Mixamo sequences for characters Goblin, Mousey, Vampire and Mremireh.

To further verify the disentanglement of our latent space, we visualize a 3D projection obtained via PCA on the latent space of the 32 pose variants for five different test poses in Figure 14. We show the clusters for our final model as well as for the ablations where parts of the disentanglement loss functions $\mathcal{L}_{\mathrm{aug}}$ and $\mathcal{L}_o$ were removed. We observe that $\mathcal{L}_{\mathrm{aug}}$ encourages disentanglement, while $\mathcal{L}_o$ pushes clusters apart.

In Figure 15, we use the same PCA projection to visualize the same sequences of various Mixamo characters, *Goblin*, *Mousey*, *Vampire*, and *Mremireh*. As shown in Figure 6, the character topologies and scales vary significantly, resulting in different poses even for the same frames. This is reflected in the traversal of the latent space, as the trajectories for different characters are offset between different characters.

### A.6 ALIGNMENT OF CONFIGURATION SPACES

To understand why previous methods require alignment of their configuration spaces, consider two toy topologies, $a$ and $b$, each consisting of a single bone. The bone offset for topology $a$ is $o_a = (1, 0, 0)$, while for topology $b$, it is $o_b = (-1, 0, 0)$. To represent the same pose, where $\mathbf{x}_a \equiv \mathbf{x}_b$, the joint rotations must satisfy $\theta_a = \{\mathbf{R}\}$ and $\theta_b = \{\mathbf{R}\mathbf{R}_{180y}\}$, where $\mathbf{R} \in \mathbb{SO}(3)$ is an arbitrary rotation, and $\mathbf{R}_{180y}$ represents a 180-degree rotation around the y-axis.

However, there are infinitely many valid solutions for aligning $\theta_b$ to $\mathbf{x}_a$, such as rotations around different axes. This ambiguity complicates the learning process, as the model must determine which specific rotation to use. The choice of the rotation plane depends on the inverse kinematics (ik) algorithm. Consistently using the same ik algorithm across all datasets ensures uniformity in the rotation subspaces, making learning-based approaches feasible.

In contrast, directly learning in Euclidean space eliminates the need for alignment between configuration spaces. Each joint is treated independently, simplifying the learning process and allowing for the inclusion of new topologies without additional alignment. This approach assumes internal consistency within each topology, significantly reducing the complexity of integrating new skeletons.

### A.7 ANCHOR REPRESENTATION

Similar to other retargeting methods such as Aberman et al. (2020), we utilize a common representation to learn human poses. However, unlike methods that operate in configuration space and thus require a minimal sub-skeleton, our approach employs a common representation, which we call the **anchor** representation, with arbitrary complexity. This anchor representation can even be a superset of many of the used skeletons. This flexibility is possible because we learn the representation from poses in Euclidean space rather than in configuration space.

Joint positions in Euclidean space, unlike local joint rotations, are global and independent of each other, meaning that they do not explicitly depend on their neighbors in the kinematic chain. We exploit this property by using the 3D joints of SMPL-H (Romero et al., 2017) as our anchor topology due to its widespread availability and its ability to represent complex poses, including detailed finger movements (see Figure 9b). Notably, we represent the anchor pose $\mathbf{x}_{\bowtie} \in \mathbb{R}^{J_{\bowtie} \times 3}$ in Euclidean space, unlike previous works.

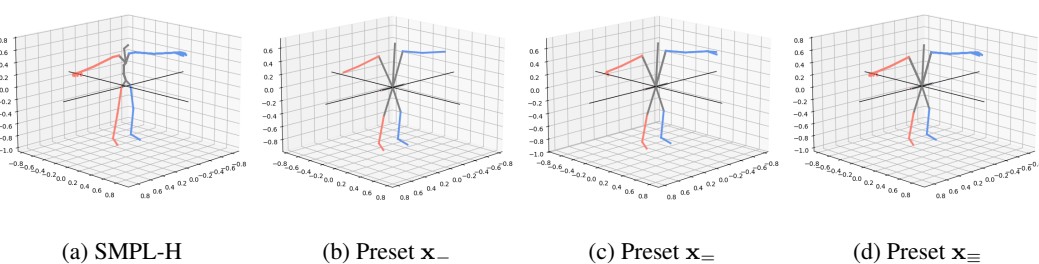

| (a) SMPL-H | (b) Preset $\mathbf{x}_-$ | (c) Preset $\mathbf{x}_=$ | (d) Preset $\mathbf{x}_\equiv$ |

Figure 16: Visualizations of normalization and presets. 16a shows a SMPL-H pose in a normalized Euclidean frame, where the hip center is at the origin and where the hip lies on the y-axis. 16b shows preset $\mathbf{x}_-$, 16c shows preset $\mathbf{x}_=$, and 16d shows preset $\mathbf{x}_=$.

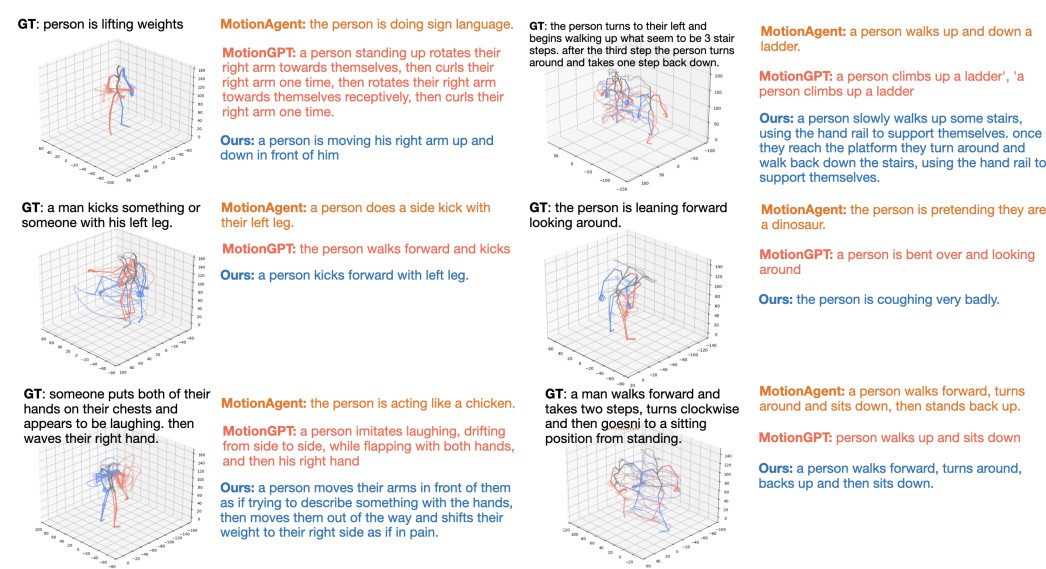

Figure 17: Sample captioning on the HumanML3D test set with MotionAgent (Wu et al., 2025), MotionGPT (Jiang et al., 2023) and our method.

The independence of each joint in Euclidean space allows us to use arbitrary subsets of $\mathbf{x}_{\bowtie}$ without the need for re-fitting the data, as required in previous methods. In this work, we define three pose subsets: $\mathbf{x}_-$, $\mathbf{x}_=$, and $\mathbf{x}_\equiv$, which represent minimal common skeletons without fingers, with two fingers (thumb and index finger), and with all five fingers, respectively. We visualize those presets in Figure 16. Hips exhibit significant variability in annotation across different human pose topologies. For instance, SMPL-based models use the head of the femur, while many professional motion capture systems, such as Vicon, model the hips further apart, resembling the Greater Trochanter of the femur (see Figure 9d). To address this, we define a pseudo-hip, positioning the origin of the reduced skeletons at the center of the two hip joints. We set the pseudo-hip width equal to the shoulder width and maintain the original direction of the pseudo-hip. Sample reduced skeletons are illustrated in Figure 4.

These reduced skeletons can be directly used to encode unseen topologies into our latent representation without the need for additional training or fine-tuning.

### A.8 CAPTIONING

In Figure 17, we visualize some captioning examples and compare them to the results from Motion-Agent (Wu et al., 2025) and MotionGPT (Jiang et al., 2023).

## A.9 LIMITATIONS

One limitation of our method is its inability to directly retarget to a different style based on a target skeleton topology. This limitation is most evident in our intra-structural retargeting experiments in Section 4.3.1, where our method underperforms compared to specialized approaches. We attribute this to two main factors: first, our method prioritizes pose fidelity over style, meaning it preserves the original source pose under the new target skeleton, as illustrated in Figure 6c; second, our approach operates strictly on a per-frame basis, whereas state-of-the-art retargeting methods leverage motion sequences to capture style.

This design choice is intentional, as our focus is on achieving high pose fidelity rather than encoding motion or style. For generative models, it is critical that the representation remains as close to the respective pose as possible, ensuring accuracy and consistency.

One way to mitigate this in the future is to learn a function which can predict the offset in latent space, as seen in Figure 15, given source and target topology.

