# OpenReview forum: "Unified Pose Embeddings: Utilizing Euclidean Space for Simplified Topology Alignment"
_ICLR.cc/2026/Conference — ICLR 2026 Conference Withdrawn Submission_

### Official Review · Reviewer_F16V · 2025-10-28

**Soundness:** 3
**Presentation:** 2
**Contribution:** 2
**Rating:** 4
**Confidence:** 3

**Summary:**

The paper addresses the long-standing challenge of representing motion across diverse human skeleton topologies in generative human motion modeling. It proposes a topology-agnostic motion representation that learns in Euclidean space rather than joint rotation space. Specifically, for each skeleton topology, joint rotation data is converted to Euclidean coordinates via forward kinematics, after which a learned projector maps the pose into a shared anchor space. An autoencoder is then used to encode the pose into a latent space and reconstruct it within the anchor skeleton. Finally, the reconstructed anchor pose is mapped back to the target topology, and a learned inverse kinematics network projects the reconstructed Euclidean pose back into the joint rotation space. The paper compares the proposed method with several baseline approaches across different applications, including motion reconstruction, motion retargeting, and motion generation, and reports consistent improvements on these tasks.

**Strengths:**

- The paper raises an interesting and timely question about how motion representation affects generalization in generative motion modeling, particularly by contrasting Euclidean-space representations with traditional joint rotation spaces.

- The proposed method is straightforward, conceptually sound, and easy to implement, making it both practical and broadly applicable.

- Extensive experiments are conducted across multiple applications—including motion reconstruction, retargeting, and motion generation—demonstrating the versatility and effectiveness of the proposed representation.

**Weaknesses:**

- Although the overall idea is straightforward, some methodological details are difficult to follow. For instance, the roles of \(x_{-}\) and \(x_{=}\) are not clearly explained, and it is unclear how these representations contribute to the overall learning pipeline.

- In the experiments, the paper compares the proposed modular framework with baseline methods such as NKN, Skeleton-Aware Networks, and SAME. However, these baselines are designed as universal models that handle multiple skeletons within a single network, whereas the proposed approach relies on modular components trained separately. This difference raises concerns about the fairness and validity of the comparisons.

- The architectural novelty of the method is limited, as it primarily combines standard components such as autoencoders and MLP-based inverse kinematics networks. While the simplicity of the design is appreciated, the paper would be stronger with deeper analysis or insight into *why* and *how* representing poses in Euclidean space leads to the observed performance improvements.

**Questions:**

- In Section 4.1, the paper studies the learned inverse kinematics results. Could the authors clarify whether there are any new technical contributions or innovations in how the inverse kinematics model is learned, beyond adopting an MLP-based approximation?

---

> ### Author Response · Authors · 2025-11-14
> **Response**
>
> Thank you very much for the detailed review.
>
> The consensus among the reviewers is that the paper is not ready for publication, and we thus withdraw our submission. Thank you very much for your valuable feedback;
>
> We highly appreciate the time that you have spent on reviewing our work and we thus want to respond to some of your comments:
>
> —
>
> Roles of presets: In Figure 16 we provide a visual representation of the presets. The presets are Euclidean-space-based joint subsets of a typical human body (hand, arms, legs, …). This way, when presented with a new unseen topology, one could directly use one of the “presets” for which pre-trained encoders exist. This way no training at all is needed for a new topology. If one would want to retarget this topology onto another KNOWN topology (for which a decoder exists), the user could directly decode into the target topology - no training at all is needed for the unknown skeleton. Only if one would want to decode into the new topology we would need to train a (minimal) decoder to recover the IK for that particular topology.
>
> Fairness: NKN, SAME, etc require pre-aligned topologies - we provide a description for this in Appendix Section A.6. Also note that only SAME can process different topologies in the same network. On top of that, SOTA methods are trained on large amounts of Mixamo animations, on which the methods are evaluated on. In contrast, our embedding space is learned on AMASS and we utilize only around 10 mins of Mixamo motion to recover the IK.
>
> Simplicity: we believe the simplicity is the methods strength: it’s easy to implement while it produces strong results. We discuss the why’s to some degree in our introduction (l085-l096), Figure 3, Section 3.1, Section 3.2 and Appendix section A.6. We hypothesize that non-Euclidean, local, hierarchically + sequentially structured data (tree structures) are difficult to learn for neural networks, both structurally and from a loss function perspective. In contrast, using global 3d coordinates (which are “functionally” independent of each other - as their values represent their global position, and not a local one, that is relative to its parents) is easier for a nn to learn + one could use simple loss functions such as MSE.
>
> IK model: we follow previous method “A neural network based approach to inverse kinematics problem for general six-axis robots: Sensors, 2022”.
>
> —
>
>
> (To respect your time we do not want to start a discussion - we will not be able to respond to your comments after withdrawing)
>
> Again, thank you so much for your valuable feedback!

---

### Official Review · Reviewer_DNnF · 2025-10-29

**Soundness:** 1
**Presentation:** 1
**Contribution:** 2
**Rating:** 2
**Confidence:** 5

**Summary:**

The paper proposes to unify the humanoid skeletal pose representation with an embedding learned from joint positions, or Euclidean space, as in the paper. Different humanoid skeletal morphologies are canonicalized to the SMPL-H skeleton ("anchor" skeleton as termed in the paper) using an off-the-shelf retargeter or mapping to preset skeletons (without fingers, with simplified fingers, and with full fingers) to learn the mappings between the anchor skeleton and the source skeletons. From the canonicalized joint positions, the method learns the latent pose space with an encoder-decoder architecture. The challenge here is to learn the latent pose representation agnostic to the skeletal proportions (phrased as "body shape" in the paper) described by the "skeleton offsets" (translations relative to the parent joint). For this disentanglement, the encoder outputs both z_o ("skeleton offset" latent) and z (pose latent). An auxiliary decoder is then trained to recover the skeleton offset given z_o. The pose decoder is also trained to recover the joint positions given the pose latent and the augmented skeleton offset latent (output from another MLP given skeleton offset). The skeleton offset augmentation randomly scales up, down, and samples skeleton offsets from other characters to encourage the pose latent to be agnostic to the skeleton offsets. The setup is numerically evaluated in errors in IK, reconstruction, and retargeting. The applications in retargeting, text-to-motion, and motion-to-text (captioning) are demonstrated.

**Strengths:**

The goal to propose a unified pose representation is ambitious.

**Weaknesses:**

* Confusing writing
  * It is fine to use words like Euclidean space / Cartesian coordinates, but essentially, all this paper is doing is learn with joint positions
  * "Body shape" in this paper is not the shape (outer skinned geometry) as in SMPL and related papers. Instead, in the context of this paper, "body shape" refers to bone lengths or body proportions
  * L264 and L267: why put loss notation next to the predicted quantity???
  * L473: "absence of global transformation" ??? FK should put joint positions in the world space, and the joint positions should capture the global transformation
* Questionable premise and insufficient discussions on its limitations
  * The premise is that joint positions are better. But the autoencoder setup tries hard to eliminate the entanglement of the joint positions and the bone lengths (identity-specific feature). Then why not use joint rotations, which are not tied to bone lengths? There are previous papers (e.g., papers by Sebastian Starke and Daniel Holden) using the combination of joint positions and rotations for the pose representations.
  * Poses/motions learned with joint positions embedded in the world space become translation and rotation-dependent. E.g., a motion translated on the ground plane should be recognized as the same motion. The proposed setup breaks this expectation.
  * Positional representation will lose the original rotation information (twist along the bone direction). IK, whether learned or not, cannot recover this twist component, as this is ambiguous.
* Insufficient visual presentation of results
  * Skeletal poses and animations should always be presented with the skinning to make sure there are no twisting artifacts (stick figures of skeletons cannot tell the twist)

**Questions:**

Please answer the questions raised in the Weaknesses.

---

> ### Author Response · Authors · 2025-11-14
> **Response**
>
> Thank you very much for the detailed review.
>
> The consensus among the reviewers is that the paper is not ready for publication, and we thus withdraw our submission. Thank you very much for your valuable feedback;
>
> We highly appreciate the time that you have spent on reviewing our work and we thus want to respond to some of your comments:
>
> —
>
> Simplicity: Simplicity is the strength of the method: it yields very strong results, while being simple and easy to replicate.
>
> L264 + L267: this is to make clear where the loss is coming from - we can see how this could be mis-understood as a function call - we will signify this in future works. Thanks for pointing this out!
>
> L473: I believe there is a misunderstanding: the clustering was done in the latent space with no global rotation and translation. We believe that adding these information could help the model produce better results (as right now, it only sees poses in a canonical frame). No FK was applied.
>
> “The premise is that joint positions are better, why not use joint rotations”: We discuss the reasons in Section 1, Figure 2, Figure 3, Section 4.2, Section 4.3.1, Appendix Section A.1, Appendix Section A.5, Appendix Section A.6 and we discuss its limitations in Appendix Section A.8.
>
> translation and rotation-dependent: This is true: to mitigate this we utilize a canonical reference frame for the Euclidean space which we briefly describe in Appendix Section A.7. This is achieved via a rigid transformation. To recover global transform, we simply utilize the inverse transformation. However, we agree that this should be discussed more clearly in the paper. Thanks for pointing this out!
>
> “lose the [of] original rotation”: This is true for any retargeting. Our experiments show that neural networks are surprisingly robust in learning IK
>
> Insufficient visual presentation: we agree and will provide better visualizations for future works. Thanks for pointing this out.
>
> —
>
>
> (To respect your time we do not want to start a discussion - we will not be able to respond to your comments after withdrawing)
>
> Again, thank you so much for your valuable feedback!

---

### Official Review · Reviewer_orpB · 2025-10-31

**Soundness:** 2
**Presentation:** 3
**Contribution:** 2
**Rating:** 4
**Confidence:** 3

**Summary:**

This paper proposes Unified Pose Embeddings (UPE), a topology-agnostic human-motion representation learned in Euclidean joint space instead of configuration space. The method uses (i) an anchor representation (SMPL-H joints) with optional reduced presets x^{-}, x^{=}, x^{\equiv} to cover datasets lacking paired anchors; (ii) a lightweight autoencoder that disentangles pose z from shape o via offset regression and shape augmentation; and (iii) a per-topology neural IK module (5-layer MLP, with optional gradient refinement) to map 3D joints back to rotations. Experiments report: strong IK accuracy across five datasets (Table 1–2), competitive retargeting on Mixamo with ~10 minutes of per-character data (Table 4, Fig. 6), text-to-motion with MDM/MARDM showing comparable recall and precision but somewhat higher FID (Table 5), and captioning results via k-means tokenization (Table 6). Claimed data efficiency: IK generalizes with ≈16k frames (~5.33 minutes at 50 Hz) for a new topology (Fig. 5).

**Strengths:**

- anchor representation + disentanglement + per-topology IK is easy to implement and extend; components can be trained independently.
- convincing curves and table ablations (linear vs MLP-2/MLP-5, with/without post-opt). The stricter accuracy metric (max-limb threshold) is thoughtful.
- x^{-}, x^{=}, x^{\equiv} reduce reliance on fragile full retargeting when anchors are unavailable.
- shape-offset regression + augmentation measurably helps reconstruction/disentanglement (Table 3; PCA analyses).

**Weaknesses:**

- Although topology-agnostic is claimed, the pipeline anchors to SMPL-H and still requires learning g(\cdot) and IK r(\cdot) per topology; cross-domain generality is thus partly deferred to new training. Evidence on unseen, highly non-human or production rigs is limited.
- On text-to-motion, recall improves but FID worsens vs native HumanML3D; attribution to conversion shift is plausible but leaves the core question—does Euclidean anchoring improve generative quality—only partially answered. Captioning trails MotionAgent/GPT on alignment (Table 6).
- Authors note intra-structural style transfer is weaker (per-frame design; pose fidelity over style). Stronger baselines (e.g., SAME, AnyTop) are only partially covered, and the “cross-structural becomes trivial in Euclidean space” risks overstating difficulty reduction without broader rigs.
- No stress tests for noisy joints, missing markers, frame-rate variance, or domain shift. The “5.33 minutes” data claim depends on 50 Hz and Human3.6M; sensitivity to sampling rate and motion diversity isn’t quantified.
- Many results ultimately traverse anchor --> target or target --> anchor paths; cumulative conversion noise is acknowledged but not bounded with diagnostics (e.g., cycle errors per joint, hand articulation fidelity).

**Questions:**

1. How far can rigs deviate (extra twist bones, non-human proportions, non-tree constraints) before the preset/anchor approach fails? Any results on production rigs beyond SMPL/H36M/LAFAN?
2. How does the “~5.33 min” generalization change with frame rate, motion diversity, or label noise? Please plot accuracy vs. minutes at 25/30/60 Hz and with missing joints.
3. Can you report anchor-native evaluation (no conversion to HumanML3D) or a conversion-robust metric (e.g., Procrustes-aligned feature FID) to isolate representation benefits?
4. What are wall-clock train times and parameter counts for g(\cdot) and r(\cdot) per new topology? Any amortization via meta-learning or shared adapters?
5. Could a temporal/style head (or latent offset predictor) mitigate intra-structural style loss without sacrificing your pose-fidelity goal (cf. your Appendix A.9)?

---

> ### Author Response · Authors · 2025-11-14
> **Response**
>
> Thank you very much for the detailed review.
>
> The consensus among the reviewers is that the paper is not ready for publication, and we thus withdraw our submission. Thank you very much for your valuable feedback;
>
> We highly appreciate the time that you have spent on reviewing our work and we thus want to respond to some of your comments:
>
> —
>
> “Topology-agnostic” —> we concede that “topology-agnostic” is an overstatement - we followed the naming schemes of previous methods (SAME, AnyTop) which themselves are not truly “agnostic” in the literal sense (i.e. require pre-alignment of every topology). However, as you have stated, requiring different encoders for different skeletons is not agnostic and we will remove this claim in future versions of our work. Thanks for pointing this out.
>
> With regard to FID: note that FID is highly dependent on the encoder. For example, recent works have shown that the HumanML3D encoder is highly biased in favor of discrete representations, which does NOT track with human judgment [Rethinking diffusion for text-driven human motion generation, CVPR25].
> In Appendix Section A4 we discuss other works that also experience an increase in FID due to converting between different representations. This is a common phenomenon not unique to our method.
>
> Intra-structure: Our main focus is on finding a representation that can be utilized as a drop-in replacement for generative motion models , but which can be easily transferred into various target representations without the need of manual retargeting - not retargeting per se. In fact, we show that our method is NOT great in character-to-character retargeting (Table 4, Intra) - nevertheless, it is capable of doing so if needed. However, our method excels at cross-topology retargeting (Table 4, Cross), where the same general body size is used, but the bone structure is changed - this is extremely challenging for methods that directly rely on the kinematic tree structure (as the tree is “different” now) but becomes “trivial”  when utilizing the 3D keypoints. This is the main insight of our paper.
>
> Stress test + Discussion on cumulative conversion: this is a great suggestion which we will add to our future work. Thanks!
>
>
> 1.) The system is astonishingly flexible to the rigs: for example, for Section 4.3.1 + Table 4 + Figure 6 we evaluate on the Mixamo dataset, which has a large variation in human shapes (and leans towards cartoon’y animations). For encoding we utilize the pre-trained preset, which has been trained on AMASS, which is not cartoon’y and instead directly taken from human captures.
> Only the last part of the network, g, and r, are fine-tuned on the Mixamo-skeleton. Nevertheless, the model can represent very cartoony characters with very unrealistic proportions, such as Mousey and Vampire.
>
> 3.) Table 3 discusses the reconstruction in terms of the anchor representation while Table 1 discusses the end-to-end reconstruction evaluation for various motion capture datasets.
>
> 4.) A shared adapted could work for $g$ (Euclidean space) but would be extremely challenging for $r$ (rotation space), as $r$ depends on the skeleton structure and bones. Methods that attempt to learn a joint adapter (like AnyTop) require to pre-align the datasets beforehand - directly utilizing the datasets “as is” does not work. We discuss the reasons behind this in Appendix section A 6. Aligning the dataset / topology beforehand is not desirable when dealing with many different and changing skeleton topologies.
>
> 5.) Yes, a temporal model, in principle, would do much better on the intra-structure retargeting task. However, we consciously chose a per-frame approach which can deal with sequences of any temporal resolution. Many temporal methods “bake in” the temporal resolution of the training data and cannot extrapolate beyond it, which we wanted to avoid. However, our representation could be used on top of a (different) temporal model which then maps poses in our latent representation to better match the intra-retargeting task.
>
> —
>
>
> (To respect your time we do not want to start a discussion - we will not be able to respond to your comments after withdrawing)
>
> Again, thank you so much for your valuable feedback!

---

### Official Review · Reviewer_sv7N · 2025-11-01

**Soundness:** 2
**Presentation:** 3
**Contribution:** 2
**Rating:** 2
**Confidence:** 4

**Summary:**

In this paper, the authors propose to use the euclidean joint positions to represent the human pose,
which is a simple and effective way to represent the human pose across different skeletons.
A set of latent encoder and decoder is used and latent space is trained to be disentangled betweenthe body shape and the pose

**Strengths:**

I really like what the paper is studying and trying to solve.
My team actually had a similar discussion about the universal motion representation across different morphologies,
and we believed "end-effectors" positions can be a potential candidate for this.
This is similar to the idea of euclidean space in the paper but the authors push the idea further by introducing a disentanglement learning strategy.

1. The paper discuss a curical problem in animation,
how to handle retargetting across different proportions and morphologies.

2. The paper provides multiple resolution of skeleton representation which is flexible to use for different applications.

3. Disentanglement is studied in the paper, which opens the door to unsupervised motion feature learning.
This will allow us to learn the features from huge ton of skeleton morphologies and proportions.

**Weaknesses:**

1. The paper does not seem finished or ready.

The result section is very weak and we are seeing very limited visual results.
There are no meshed character visualized at all except for figure 1, which is also not very informative.
It's hard to draw any conclusion from the results.

2. The scalability of the number of characters or morphologies is not well studied.
A small subsets of characters of in mixamo (4 i believe?) is not enough to make a general statement about the scalability for the proposed method.

3. The algorithm does not consider character mesh, which is crucial to consider retargetting.
This is central in industry application and was briefly studied in [1, 2].
This needs to be studied before this retargeter can be useful for industry applications.

[1] Ho, Edmond SL, Taku Komura, and Chiew-Lan Tai. "Spatial relationship preserving character motion adaptation."
In ACM SIGGRAPH 2010 papers, pp. 1-8. 2010.
[2] Yang, Lujie, Xiaoyu Huang, Zhen Wu, Angjoo Kanazawa, Pieter Abbeel, Carmelo Sferrazza, C. Karen Liu, Rocky Duan, and Guanya Shi.
"OmniRetarget: Interaction-Preserving Data Generation for Humanoid Whole-Body Loco-Manipulation and Scene Interaction." arXiv preprint arXiv:2509.26633 (2025).

4. There's no studied about keeping the semantic meaning of the motion.
For example a slow walking might be retargeted into fast walking for a smaller person,
which is common in traditional retargetting methods.

**Questions:**

please refer to the above section.
I think the project is going towards the right direction, but it's far from being ready to be viewed as a complete project.

---

> ### Author Response · Authors · 2025-11-14
> **Response**
>
> Thank you very much for the detailed review.
>
> The consensus among the reviewers is that the paper is not ready for publication, and we thus withdraw our submission. Thank you very much for your valuable feedback;
>
> We highly appreciate the time that you have spent on reviewing our work and we thus want to respond to some of your comments:
>
> —
>
> 1.) This is an oversight on our part - we will provide better visualizations in future submissions.
>
> 2.) This experiment follows existing benchmarks to compare to SOTA
>
> 3.) That is true and an interesting future direction: however, there is a lack of publicly available articulated mesh  datasets which makes this tricky. For example, while Mixamo has many different skeleton (and mesh) topologies - However, those are still, fundamentally, derived from the same base skeleton - and furthermore utilize the same IK algorithm. This “masks” the problem that many SOTA methods (AnyTop, SAME) still require all datasets to be “aligned” structurally - or have to be aligned in a dataset pre-processing step, i.e. with MotionBuilder - This makes adding new skeleton topologies difficult, as they first have to be pre-processed. By directly learning in the provided data space we completely work around this constraint and are able to integrate any humanoid skeleton, requiring only a limited number of training sequences to learn the given new topology.
>
> 4.) Our main focus is on finding a representation that can be utilized as a drop-in replacement for generative motion models , but which can be easily transferred into various target representations without the need of manual retargeting - not retargeting per se. In fact, we show that our method is NOT great in character-to-character retargeting (Table 4, Intra) - nevertheless, it is capable of doing so if needed. However, our method excels at cross-topology retargeting (Table 4, Cross), where the same general body size is used, but the bone structure is changed - this is extremely challenging for methods that directly rely on the kinematic tree structure (as the tree is “different” now) but becomes “trivial”  when utilizing the 3D keypoints. This is the main insight of our paper.
>
> —
>
>
> (To respect your time we do not want to start a discussion - we will not be able to respond to your comments after withdrawing)
>
> Again, thank you so much for your valuable feedback!

---

### Author Response · Authors · 2025-11-14
**Thank you - Withdrawal**

The consensus among the reviewers is that the paper is not ready for publication, and we thus withdraw our submission. We thank all the reviewers for the time that they took to read and critic our work.

We will individually respond to each reviewer to answer their questions and comment on some of the criticisms. In appreciation of everyone’s time we do NOT want to start a discussion (as we are withdrawing the paper and will not be able to comment anyway).

---

### Note · Authors · 2025-11-14

**Comment:**

The consensus among the reviewers is that the paper is not ready for publication, and we thus withdraw our submission. We thank all the reviewers for the time that they took to read and critic our work.

We individually responded to each reviewer to answer their questions and comment. We highly appreciate the valuable feedback that we have received.

**Withdrawal Confirmation:**

I have read and agree with the venue's withdrawal policy on behalf of myself and my co-authors.